# Photochemical Water Splitting via Transition Metal Oxides

**Fateh Mikaeili**, **Tessa Gilmore \* and Pelagia-Iren Gouma**

Department of Material Science and Engineering, The Ohio State University, Columbus, OH 43210, USA
\* Correspondence: gilmore.281@osu.edu

**Abstract:** Rapid population growth and ever-increasing energy consumption have resulted in increased environmental pollution and energy demands in recent years. Accordingly, studies and research on innovative and efficient ways of wastewater clean-up and exploiting eco-friendly and renewable energy sources such as sunlight have become a necessity. This review focuses on recent progress with photocatalysis for water splitting capabilities. It introduces photocatalysis and hydrogen as a fuel source, before moving on to explain water splitting. Then, the criteria for ideal photocatalytic materials are discussed along with current material systems and their limitations. Finally, it concludes on the TiO$_2$ systems and their potential in future photocatalysis research.

**Keywords:** photocatalysis; photochemical; water splitting; hydrogen; TiO$_2$

## 1. Introduction to Photocatalysis

Within one hour, the sun provides more energy to the surface of the earth than the human population needs for one year. Considering the ever-increasing energy demands and concerning environmental issues, the pursuit of such an enormous source of eco-friendly and renewable energy is more important now than ever [1,2]. Therefore, a significant number of current projects have been dedicated to tackling the challenges of making chemical fuel from sunlight.

Solar-to-chemical energy conversion provides a beneficial way to store solar energy that is both sustainable and efficient. The question then arises as to how we split water molecules in order to generate hydrogen gas for fuel. The first instance of water splitting through hydrolysis was reported around 1800, and since then numerous studies of alternative ways to achieve this reaction were reported. Unfortunately, the capital cost of mining hydrogen from water remained higher than the present-day industrial process of methane steam reforming [3]. However, with the increased pressure from government and independent groups for renewable and efficient energy, hydrogen generation through water splitting has been given more focus.

When water comes into contact with sunlight within a photocatalyst platform, it dissociates to produce a clean energy source of hydrogen without further energy requirements from the photocatalytic process [4,5]. Researchers have suggested and investigated various systems for this photocatalytic water splitting process [6], some of which will be discussed in this review. The fundamental elements related to efficient hydrogen generation through one-step photochemical water splitting are first examined. Then, the essential steps of a photochemical reaction, along with chemical, electronic, and physical criteria for the selection of photocatalytic material, and an overview of the current material systems is reviewed. Finally, this review will discuss the TiO$_2$ systems and their potential in further photocatalytic research.

## 2. Hydrogen as Fuel and Its Production

The smallest and lightest element in the universe is Hydrogen. Its unique chemical and physical properties offer both advantages and challenges in developing it as a prevalent fuel source. The uniqueness of hydrogen compared to other chemical fuels comes from its

high diffusivity, inimitability, combustibility, very low viscosity, and electrochemical properties [7]. All of these characteristics together make hydrogen different or more favorable than other gaseous fuels. Moreover, it has been reported that hydrogen engines perform more efficiently than gasoline engines. This is due to the low autoignition temperature of hydrogen as well as the higher octane rating compared to conventional gasoline. Hydrogen also has the highest heating value, which is 52,000 Btu/lb, among all the available fuels [8].

Fossil fuels are limited in the ways they can be mined or extracted. The advantage of hydrogen is its abundant different sources such as oil, coal, natural gas (NG), biomass, water, etc. Currently, the primary source of hydrogen is NG due to the low production cost and high efficiency. Figure 1 below illustrates the current most notable sources of hydrogen for industrial use. Hydrogen is largely generated from NG (48%), raw petroleum products (30%), coal (18%), and electrolysis of water (4%) [9].

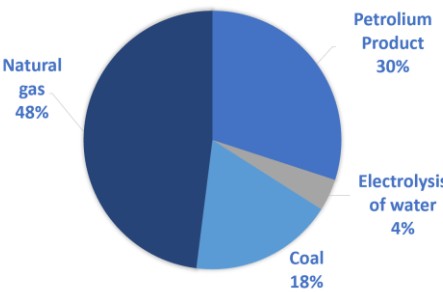

**Figure 1.** Percentage of production of hydrogen from petroleum products, natural gases, coal, and electrolysis of water. Values taken from [9].

Almost 96% of today's industrial hydrogen production is sourced from fossil fuels. In truth, the current hydrogen fuel is simply relying on another byproduct of non-renewable fuel sources and would ultimately cost more than any of the initial forms of the fossil fuels themselves. However, petroleum supplies will become limited in the near future [10]; therefore, it is critical for renewable hydrogen sources to gain attention and support.

## 3. Water Splitting

As mentioned in the previous section, water could be used as the main source of hydrogen production. By definition, the dissociation of the water molecules into their constituents (hydrogen and oxygen) is known as water splitting (Reaction 1). This phenomenon was first observed around the 1860s through water electrolysis, where an electric current was passed through water to complete the water-splitting reaction. Production of hydrogen from water is highly energy-demanding. There are many different proposed systems to use other renewable energy sources, such as hydropower, wind turbines, photovoltaic cells, and so on, to provide the electrical energy required for water hydrolysis. However, the electricity that is consumed is more valuable than the hydrogen that is produced. To fully understand the energy requirements for the water-splitting reaction, the thermodynamics and electrochemistry of the process will be discussed [11].

$$2H_2O \rightarrow 2H_2 + O_2 \qquad (1)$$

### 3.1. Thermodynamics and Electrochemistry of Water Splitting

As previously mentioned, dissociation of a water molecule is a reaction that is energetically uphill. Typically, and without considering the overpotential, it requires 1.23 V per one mole of water for the complete splitting of water into oxygen and hydrogen. Due to the very low ionization power (Kw = $1.0 \times 10^{-14}$), water splitting becomes thermodynamically unfavorable (Gibbs free energy LG0 = 237 kJ/mol, 2.46 eV per molecule) at standard tem-

perature and pressure [9]. LG is calculated at 25 °C using the thermodynamic parameters ($\Delta H$, $\Delta T$, and $\Delta S$) required for the water-splitting process:

$$\Delta G = \Delta H - T\Delta S = 285.83 \text{ kJ} - 48.7 \text{ kJ} = 237.13 \text{ kJ} \tag{2}$$

With the obtained value of the Gibbs free energy and using the help of the Nernst Equation, the standard cell potential E° of the reaction could be obtained (Equation (3)).

$$E° = -\frac{\Delta G°}{zF}. \tag{3}$$

In the above equation, z is the number of electrons transferred in the reaction (in this case, two), and F is a proportionality constant and is in the Faraday units (96,485 C/mol). Using the Nernst equation mentioned above, the standard potential of the water electrolysis can be calculated as 1.229 V at 25 °C. This cell potential belongs to the difference in potentials of the two half-cell reactions occurring at the cathode (reduction; hydrogen evolution reaction (HER)) and anode (oxidation; oxygen evolution reaction (OER)). The Nernst equations for the half-cell reactions of water splitting are mentioned below:

$$\text{Anode (oxidation): } 4OH^- + 4h^+ \rightarrow O_2 + 2H_2O \quad E = 1.23 \text{ V versus NHE} \tag{4}$$

$$\text{Cathode (reduction): } 2H_2O + 2e^- \rightarrow H_2 + 2OH^- \quad E = 0.00 \text{ V versus NHE.} \tag{5}$$

Leading to an overall reaction of the following:

$$2H_2O(l) + 4e^- + 4h^+ \rightarrow O_2(g) + 2H_2(g) \quad E = 1.23 \text{ V} \tag{6}$$

where NHE is the normal hydrogen electrode.

### 3.2. Solar Water Splitting

The above calculations illustrate how thermodynamically unfavorable this reaction is; however, nature shows us that it has been done for millions of years in plant leaves through photosynthesis—a process in which the plant splits absorbed water using sunlight and transforms it into fuels in the form of hydrocarbons. Therefore, a reasonable platform would be the combination of solar energy with the plentiful water resources available to us. This platform is called solar water splitting and is generally completed by photobiological, thermochemical, or photocatalytic water splitting. Table 1 below contains a summary of each method.

**Table 1.** Summary of solar water splitting methods.

| Solar Water Splitting Methods | Description | Comments |
|---|---|---|
| Thermochemical | Uses high temperature—from concentrated solar power and chemical reactions. | The simplest method. The requirements for large solar concentrators make this method very expensive [12]. |
| Photobiological | A process in which light is used in biological systems to dissociate water into molecular oxygen and hydrogen. | Low yields of hydrogen production, toxic effects of enzymes, limitations on scaling up [13]. |
| Photocatalytic | Theoretically, only light energy, water, and catalyst are needed. | Low cost, relatively higher solar $H_2$ efficiency, the capability of separating $H_2$ and $O_2$, and flexible reactor size. |

## 4. Photocatalytic Water Splitting

Photocatalytic water splitting uses sunlight, water, and a semiconducting photocatalyst to dissociate water molecules through the two redox reactions mentioned above. The breakthrough study was started in 1972 by Fujishima and Honda [14] in a photoelectrochemical cell using $TiO_2$ as their photocatalyst. Afterwards, photocatalytic water splitting received an enormous amount of attention due to its potential. During the past 40 years, various photocatalyst materials and systems were used to split water under ultraviolet light or visible light. Photocatalytic water splitting could be categorized into either photochemical water splitting or photoelectrochemical (PEC) water splitting [15].

The schematics of the two different systems are summarized in Figures 2 and 3 below. Both types include three basic steps: a semiconductor photocatalyst absorbs more photon energy than the band gap energy of the photocatalyst and excites the electron–hole pairs; the photogenerated charge carriers are then separated out and move toward different sites of the photocatalyst's surface; finally, at these sites water reacts with the charge carriers in two separate redox reactions and therefore is reduced by electrons to produce $H_2$ in the same time its oxidized by holes to produce $O_2$ [16].

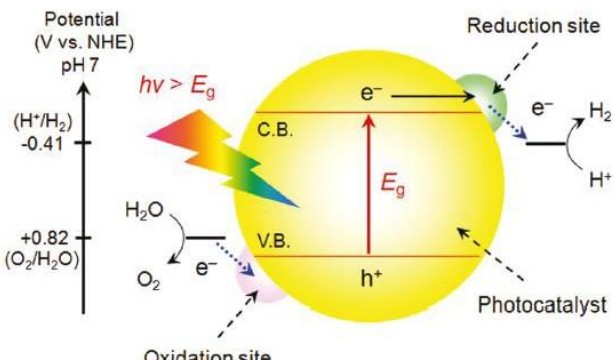

**Figure 2.** Conventional one-step system (photochemical water splitting). Reprinted with permission from [16]. Copyright 2022 American Chemical Society.

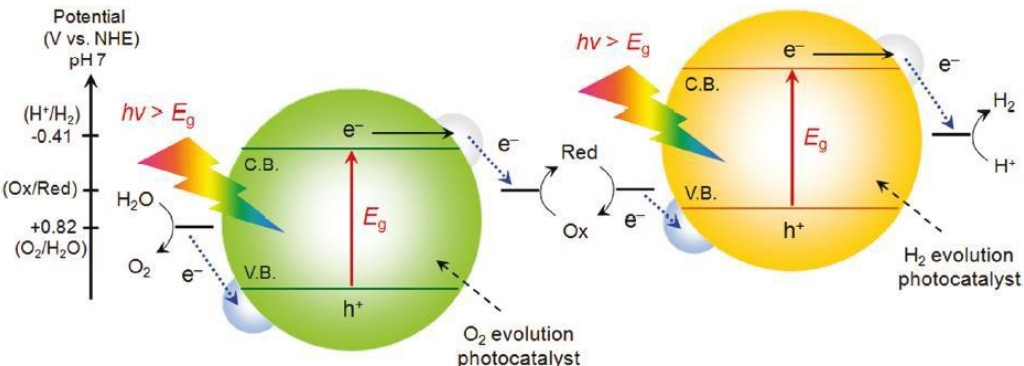

**Figure 3.** Two-step photoexcitation system with two photocatalysts in tandem (PEC water splitting). Reprinted with permission from [16]. Copyright 2022 American Chemical Society.

Even though the general concept of photochemical and PEC systems is the same, the setup is different. In photochemical reactions, the water-splitting reaction takes place at the semiconductor–electrolyte junction, whereas in a PEC setup the reaction takes place at two different sites. In this method, illuminating the cathode or anode would provide the required potential [15].

In order to understand the difference between these two setups, an important characteristic of the semiconductor should be taken into consideration. This characteristic is the band edge position of the semiconductor [17]. A suitable semiconductor for water splitting has a valance band position that is more positive than the $O_2/H_2O$ energy level

(1.23–0.059 pH, V versus NHE) and a conduction band position that is more negative than the $H^+/H_2$ energy level (0–0.059 pH, V versus NHE). In other words, in the ideal case, a single semiconductor material should have a band gap that is large enough to split water, so the conduction band energy and valance band energy should straddle the electrochemical potentials E0 ($H^+/H_2$) and E0 ($O_2/H_2O$) [18]. However, in the case of a single semiconducting material, the second requirement is not satisfied in most of the material systems (as will be discussed in depth in further sections). Scaife [19] mentioned in 1980 that it is exceptionally difficult to find a single semiconductor photocatalyst with both characteristics. This difficulty is why many studies focus on two semiconductor photocatalytic systems (PEC water splitting). By using two different materials, each one will act as either a photoanode or photocathode, which when used in tandem, satisfies the band gab requirement.

As seen in Figure 3, a two-step PEC system involves water splitting in two parts: one for the hydrogen evolution using a semiconductor that satisfies the conduction band position for that reaction, and the other for the $O_2$ evolution. In this method, a semiconductor that only partially satisfies the band edge position for the redox reaction could still be used in conjunction with another semiconductor to facilitate a water-splitting reaction [20]. However, there are many drawbacks to and critics of this method.

First, it requires the number of photons to be double that for the one-step system to achieve overall water splitting. The number of photons required in two-step photocatalytic water splitting is eight, whereas in one-step overall water splitting it is four. This difference causes the amount of hydrogen and oxygen produced in a two-step process to be half that of the one-step process at light absorption values and apparent quantum yield of unity [21].

Second, there are still some drawbacks that involve promoting electron transfer between two semiconductors and opposing and suppressing the possible backward reactions that involve shuttle redox mediators [22]. Therefore, since the number of backward electron-transfer routes increases, which is the result of an increase in the number of elementary steps, this route is kinetically unfavorable. Overall, two different studies summarized the techno-economic analyses which determined that a high capital cost prevents PEC devices from being implemented into solar hydrogen production. For these reasons, we will focus on one-step photochemical water-splitting systems.

## 5. Photochemical Water Splitting

In order to understand the photocatalytic process in photochemical water splitting (Figure 4), which is a quite complicated process, a simple step-by-step description of the process will be discussed in this section. Overall, the following steps occur:

1. Photon absorption: Photocatalysts absorb photons and generate electrons and holes at the surface. When the material absorbs the photons with an energy that is equal to or more than the band gap energy of the semiconductor, an electron jumps from the valance band to the conduction band, leaving a hole in the valance band. Electrons and holes release energy (heat) and move the conduction and valance bands to the minimum and the maximum positions, respectively [23].

2. Charge transport: After the charge carriers have been excited, there are different scenarios that could occur. The first scenario, which is highly unfavorable, is that excited-state conduction band electrons and valance band holes recombine. In the case where there is not a suitable force to separate these charged carriers, the energy stored in them will dissipate in a very short time (typically a couple of nanoseconds) in the form of heat. In the event that there is a defect state, a trap on the surface of the material, or a suitable scavenger, the recombination is potentially avoided. Another scenario is that the excited electrons and holes will move to the respective reaction sites [24]. In bulk, if the carriers do not recombine, it is only possible for either an electron or hole to be accumulated at the anode or cathode, whereas in a nanostructured semiconductor both of the photogenerated charge carriers could be present at the same surface. Low dimensionality, few numbers of defects, and a high

surface area in nanostructured materials result in key differences in electron transport compared to their bulk counterparts [25,26].

3.  After the charges have moved to the reaction sites in the material/water interface, they can participate in the surface chemical reactions between these carriers and the compounds (e.g., water) [15].

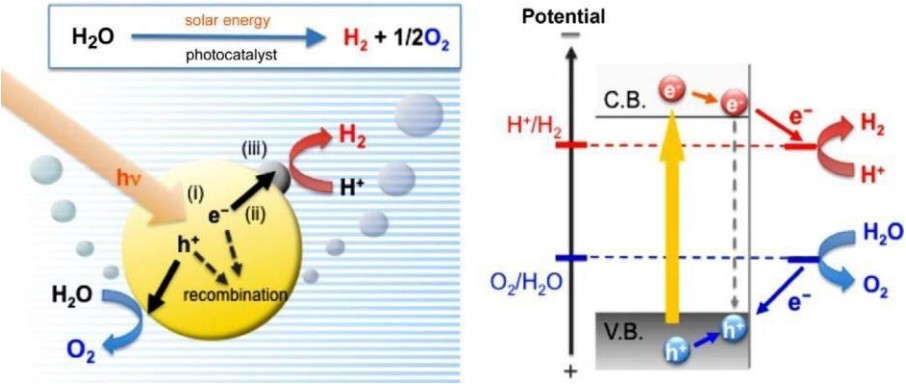

**Figure 4.** Schematics of photochemical water splitting [27]. Figure reused with permission from https://pubs.acs.org/doi/10.1021/cr1002326 (accessed on 1 September 2022); further permission related to this article should be directed to ACS.

## 6. Criteria for Selection and Synthesis of an Ideal Photocatalytic Material

As mentioned in the previous section, there are many different factors and steps in the photochemical water-splitting process. Therefore, the development of photocatalytic materials that can maximize the conversion of solar energy to solar fuel demands significant consideration. At the molecular level, several physicochemical functions need to be integrated into one stable chemical system that can set the criteria, which must be satisfied simultaneously.

### 6.1. Chemical and Electronic Properties of the Materials

1.  The band gap of the semiconducting material should lie between 1.6 eV (1.23 eV + overpotential) and 2.7 eV (larger than 2.43 eV). It is known that only 4% of the sunlight on earth is UV. Thus, in order to achieve maximum efficiency, the material should be able to harvest within the visible light spectrum [28].

2.  Band edge positions mean that band edges must straddle between the redox potentials of $H_2O$ (0.00 eV and 1.23 eV), as illustrated in Figure 4. Semiconductor materials must satisfy the minimum band gap requirement (~1.4 eV) [29]. Although recently a couple of different ways of determining the valance band edge position of new material systems have been proposed, the task of predicting the exact band position of a new photocatalyst is arduous. For this reason, the applicability of a specific photocatalyst material for overall water splitting is unknown in advance [30,31].

3.  Charge transfer is necessary at the photocatalytic surface and must be fast enough to prevent photo-corrosion and shifting of the band edges. This prevention avoids the recombination of the charge carriers and can further provide efficient oxidation and reduction sites on the surface of the material. A prevalent approach to hinder the recombination process is the use of a cocatalyst. Cocatalysts have been used in many investigations on hydrogen generation; however, some cocatalysts are highly active and induce a reverse reaction, i.e., the generation of water from molecular oxygen and hydrogen, which must be reduced [17].

4.  The stability of the material in an aqueous medium is an essential requirement (at least for 20 years). Many investigated material systems are susceptible to photo-corrosion such as CdS. Although high efficiencies are reported in the early stages of experiments,

due to oxidation of the material the process deteriorates over time before eventually stopping [32].

*6.2. Physical and Crystal Structure Properties of the Material*

Aside from the electronic and chemical properties that were mentioned above, there are structural factors of semiconductors that play a significant role in the final efficiency of the photochemical water-splitting process. Recent capabilities in advanced characterization enabled material scientists to investigate these factors, some of which are briefly discussed below:

1.  Crystalline phases: Semiconductors with different polymorphs have been shown to have different photocatalytic overall water-splitting features. One famous example of this is $TiO_2$, which has the three main polymorphs, anatase, rutile, and brookite [33,34]. Initially, it was proposed that overall water splitting could only occur using the rutile phase [35,36]. However, it was later discovered that the reaction is also feasible when anatase or brookite are treated with continued UV irradiation [37]. Early studies using infrared absorption–excitation energy scanning spectroscopy revealed that there are numerous trapped states close to the valance bands of anatase and brookite but not rutile. Later, it was concluded that the elongated emission of UV on these two phases promotes the phase transformation to a quasi-rutile structure that elevates water-splitting reactions. On the other hand, a recent experiment by the Akira group in 2017 [38] illustrated the opposite conclusion, that brookite should have better photocatalytic activities, by studying the depth of electron traps in the three different polymorphs. The group concluded that brookite has moderate trap levels that both preserves the reactivity of electrons and hinders the recombination process of the charge carriers. These inconsistent results and conclusions show that there is a lack of fundamental understanding of the different behavior of the photocatalytic activity of a semiconductor with different polymorphs.

2.  Crystallinity: A semiconductor with a high degree of crystallization has fewer structural imperfections such as vacancies and dislocations. These defects are known to be recombination sites for photogenerated charge carriers. Therefore, it is commonly suggested that a highly crystalline semiconductor has a lower electron–hole pair recombination rate [39]. However, in two distinct studies in 2019 published in the *Nature Materials* journal, two different conclusions have been derived. Wang et al. [40] studied the overall water splitting of single-crystal $Ta_3N_5$ nanorods with no surface defects and grain boundaries and compared the water-splitting efficiency of the same material system with different nanorods with relatively more defects. Their result demonstrates that the hydrogen production rate is almost three times higher in the case of nanowires with no defects. In another study by Li et al. in 2019 [41], N-doped titania samples with differing amounts of oxygen vacancies on the surface were studied. From this experiment, it was demonstrated that by increasing the oxygen vacancies on the surface by close to 13 times, these oxygen vacancies acted as electron traps and increased the final efficiency of hydrogen evolution by nearly two times. These two different conclusions show that, although it is known that defects act as recombination sites, it is possible that different types of defects could have different effects in the lifetime of photogenerated carriers that needs further investigation.

3.  Particle size and morphology: As one might expect, the particle size and morphology of the photocatalyst can have profound effects on the performance of the material [42]. In the case of particle size, the literature suggests that there is a degree of compromise that occurs [43]. That is, small particle sizes facilitate the diffusion of electron and hole pairs to the surface of the semiconductor, which results in a lower recombination rate. However, when the particle is too small, insufficient absorption of light can occur due to a lower surface area [44]. Therefore, for any particular material system, it is critical to optimize the effect of particle size. Morphology has a significant effect on determining the anisotropic distribution of charge within a semiconductor and

the overall efficiency of the water-splitting process. For example, La-doped $NaTaO_3$ demonstrated the presence of different spatially resolved reaction sites where the edges and grooves acted as reduction and oxidation sites, respectively [45]. Unfortunately, there still exists a number of studies which do not specifically address the effect of morphology on the performance, such as of ultra-aspect ratio nanofibers and nanoribbons.

4.  Heterostructures and cocatalysts: As mentioned previously, the cocatalyst is another important component that is widely used in most of the literature on overall water splitting in a single system. The literature states that the main purpose of the cocatalyst is to provide redox reaction sites. Theoretically, a cocatalyst should allow the photoexcited electrons and holes to have a smooth migration towards the surface and the reaction sites. This theory comes from the model of semiconductor and conductor interfaces that generate a built-in electric field, which should promote interfacial charge transfer [10,46,47]. However, this understanding and model is not experimentally supported, and the results do not align with the current understanding of the system. Even so, since an increased efficiency of overall water splitting in most of the material system has been reported, it is now used in ongoing research in this field [17]. On the other hand, these active sites could also promote the reverse reaction of the hydrogen evolution reaction and oxygen evolution reaction, and these reverse reactions are usually preferred to the forward reaction on a thermodynamic basis. Therefore, the use of cocatalysts in studies that require a fundamental understanding of this research area should be carried out with care. Ultimately, due to the nature of the nanostructures, uniform deposition of the cocatalysts on the materials is not usually achieved, which makes systematic studies difficult.

## 7. Current Material Systems and Limitations

As McKone and Lewis point out in their review of the ideal photocatalysts [48], regardless of the chosen solar fuel systems, certain constraints are present for their further development. These constraints arise from the three key system requirements: efficiency, stability, and scalability, which are summarized in Figure 5. Hundreds of metal oxides, nitride, and sulfide, with the electronic configuration of d0 and d10 metal cations, were tested for water splitting. Other than these materials, perovskite materials that are formed by group I and group II metals, along with some lanthanides, can also be used to catalyze photochemical water splitting.

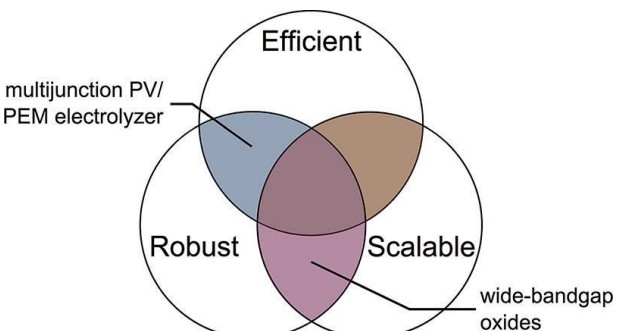

**Figure 5.** Venn diagram depicting the three requirements for an ideal solar water-splitting system [48]. Figure reused with permission from https://pubs.acs.org/doi/10.1021/cm4021518 (accessed on 5 September 2022); further permission related to this article should be directed to ACS.

As mentioned before, the key requirement for a one-step water-splitting catalyst is how the CB edge and VB edge should straddle the $H^+/H_2$ and $O_2/H_2O$ redox potentials. Figure 6 represents different semiconductors with respect to the relative position of their band edges. Furthermore, in order to be in the center of the Venn diagram above, the following necessities must be met: the catalyst system should be made from one of the

earth-abundant materials, a scalable synthesis method should be present, the system should be robust and susceptible to photo-corrosion in water, and the system should have sufficient carrier mobility. Then, in order to increase efficiency, different strategies can be implemented, such as band gap engineering to reduce the band gap and make the photocatalyst activated by visible light; morphological advancement at the surface level that improves the shape and size of the particle; and different methods of synthesis.

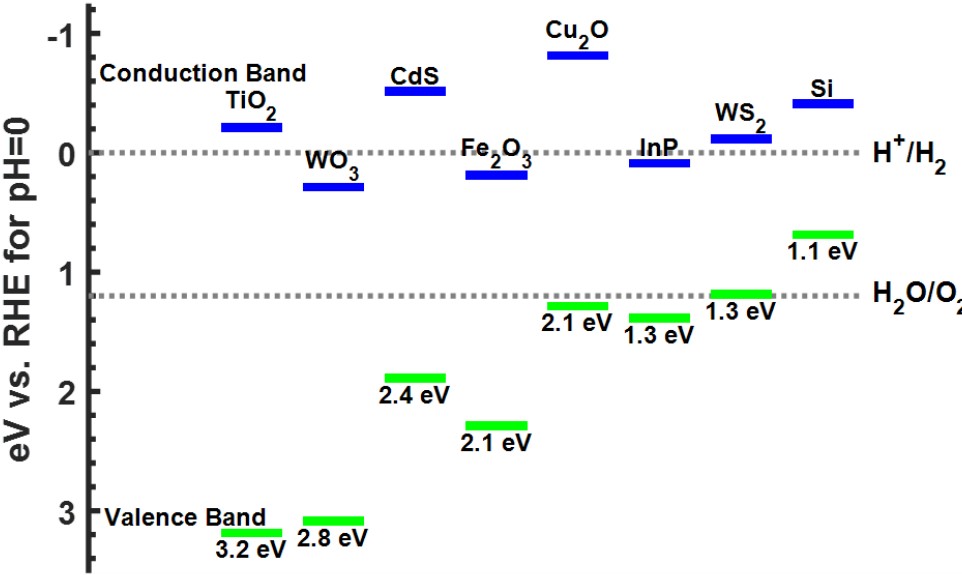

**Figure 6.** Band edge position for different material systems proposed for water splitting, where RHE is reversible hydrogen electrode, and band gap potentials are shown. Values from [49].

Based on Figure 6 and Table 2 below, most of the options do not meet the primary requirements of the band edge positions for single-step photocatalytic water splitting. A couple of remainder candidates, such as CdS, that depict high efficiencies in many studies suffer from instability in water and oxidizing shortly after catalyzing the water-splitting reaction.

**Table 2.** List of recent one-step photochemical water-splitting materials.

| Semiconductor (Available Wavelength) | Cocatalyst | | Light Source | Reactant Solution | Efficiency | Refs |
|---|---|---|---|---|---|---|
| | | Ultraviolet Light | | | | |
| $TiO_2$ (<385 nm) | Pt, $RuO_2$ | - | 450 W Xe lamp (>300 nm) | $H_2O$ with pH adjustment | AQY [1]: 30% at 360 nm | [50] |
| $NaTaO_3$:La (<300 nm) | 0.2 wt% NiO | - | 400 W Hg lamp | $H_2O$ | AQY: 56% at 270 nm | [45] |
| | | Visible Light | | | | |
| TaON (<495 nm) | 3 wt% RuOx/2.5 wt% $Cr_2O_3$-4 wt% $IrO_2$ | $ZrO_2$ | 450 W Hg lamp (>400 nm) | $H_2O$ | AQY: 0.1% at 420 nm | [51] |
| $Cu_2O$ 2.0 (<620 nm) | 3 wt% $IrO_2$ | - | 300 W Xe lamp (>440 nm) | $H_2O$ | AQY: 0.3% at 550 nm | [52] |
| CoO (<515 nm) | - | - | AMI.5G solar simulator | $H_2O$ | STH: 5% | [53] |
| $C_3N_4$ (<442 nm) | 3 wt% Pt-1 wt% CoOx | - | 300 W Xe lamp (>420 nm) | $H_2O$ | AQY: 0.3% at 550 nm | [54] |

[1] Apparent quantum yield.

The rest of the recently discovered/studied photochemical water-splitting examples mentioned in Table 2 are all suffering from low efficiencies under visible light. Among all the materials mentioned above, $TiO_2$ is still studied due to its favorable conditions. However, due to its large band gap, the pure form of titania only activates in the presence of

UV light. Although many studies illustrated high efficiencies when $TiO_2$ is used under UV light, the attempts to achieve similar results when the material is transformed into visible-light-active material through band gap engineering remained inefficient. The second barrier that is widely mentioned is titania's low carrier mobility that leads to a fast recombination rate of the charged carriers.

## 8. $TiO_2$ Systems

The crystallographic structure of $TiO_2$ is shown in Figure 7 and listed in Table 3. Well-known phases of $TiO_2$ are rutile, anatase, and brookite. Rutile is a tetragonal (a = 4.594 Å, c = 2.958 Å), anatase is also a tetragonal (a = 3.785 Å, c = 9.514 Å), and brookite is orthorhombic (a = 9.184 Å, b = 5.447 Å, c = 5.145 Å). The c coordinate of anatase is higher than other phases, and the c/a ratio of its unit cell is greater than the rutile and brookite phases. Among all three phases, anatase is the most active allotropic in terms of photocatalytic activity when compared to rutile, brookite, and $TiO_2$-B (artificial phase).

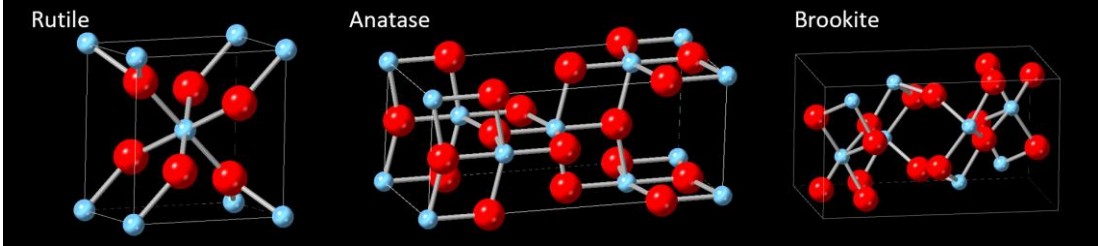

**Figure 7.** Crystallographic structures of $TiO_2$ phases.

**Table 3.** Physical properties of $TiO_2$ phases.

| Structure | Lattice Constant (Å) | | | System |
|---|---|---|---|---|
| | a | b | c | |
| Rutile | 4.594 | 4.594 | 2.958 | Tetragonal |
| Anatase | 3.785 | 3.785 | 9.514 | Tetragonal |
| Brookite | 9.184 | 5.447 | 5.145 | Orthorhombic |

Rutile is thermodynamically stable at ambient conditions. Anatase is kinetically stable and transforms to rutile at higher temperatures depending on the particle size, ambient pressure, and other parameters [55]. The brookite phase is also metastable and difficult to synthesize, and so it is seldom studied. Table 4 lists the oxide polymorphs of $TiO_2$ that have been synthesized at the nanoscale, along with the synthesis techniques, processing conditions, and relative particle size of the transforming material [56]. If the particle sizes of the three crystalline phases are equal, anatase is the most stable at sizes less than 11 nm thermodynamically, brookite is the most stable between 11–35 nm, and rutile is the most stable at sizes greater than 35 nm. These values are summarized in Table 4.

For bulk $TiO_2$, the rutile phase is thought to be a more stable phase than the anatase at room temperature. However, Zhang et al. [57] has suggested, and Gouma et al. has confirmed [58], that the total free energy of rutile is higher than anatase when the particle size is smaller than the critical size of 14 nm; therefore, anatase becomes a more stable phase. After that, some studies have claimed that rutile and anatase together have higher photocatalytic activity than pure anatase. For example, the commercial $TiO_2$, Degussa P25, is a mixture of anatase (~75%) and rutile (~25%) phases, and it is the most widely used photocatalyst for environmental applications.

Anatase can be synthesized at lower temperatures (400 °C) to nanoscale size; it shows better photocatalytic properties because nanoscale size increases the number of pores and enhances solid–solid interactions. Conversely, at higher temperatures (above 300 °C), nano-$TiO_2$ has worse photocatalytic activity.

The properties of brookite are poorly known due to the difficulties of obtaining it as a pure phase [59]. Qiuling et al. [60] also shows that the synthesized two phases (anatase/brookite) have higher photocatalytic activity due to the electron transfer from brookite CB to anatase CB, resulting in effective electron–hole separation.

**Table 4.** Different phases of $TiO_2$.

| Phase | Conditions | Particle Size (nm) | Techniques | Refs |
|---|---|---|---|---|
| Brookite | - | 11–35 nm | sol-gel | [61] |
| Anatase | room temperature | <11 nm | sol-gel | [62] |
| Rutile | >850 °C | >35 nm | sol-gel | [63] |

Two principal factors mentioned before must be taken into account when designing systems for conducting photocatalytic reactions. The first is the efficiency of $e^-$ and $h^+$ pair generation when absorbing solar irradiation. The second is the efficiency of the $e^-/h^+$ pair separation before they lose their redox ability due to recombination. In order to enhance the photocatalytic efficiency, the main issues that should be taken care of are the increase in the charge separation and recombination lifetimes of charge carriers, increase in solar spectrum response range, and changing of the selectivity or yielding of a particular product.

The efficiency of the photocatalytic process, the stability under light, the selectivity of the product, and the activation range of the wavelength are factors in determining the right aim of semiconductor photocatalysis systems. For the $TiO_2$ system, the primary barrier remains to be its activation by ultraviolet light (12' 400 nm) due to the wide band gap of $TiO_2$ (~3.2 eV for anatase and ~3.0 eV for rutile), for which it only absorbs UV light. Another major drawback of $TiO_2$ is photogenerated electron–hole recombination, which deteriorates photocatalytic activity [64]. Therefore, an obvious goal is reducing the band gap of $TiO_2$ in order to shift the absorption band to the visible region and to enhance the electron–hole separation process. Modifications of $TiO_2$ have been accomplished by different strategies such as doping, cooping, sensitization, and coupling. The coupling of two semiconductors provides different energy levels, which can allow a more efficient charge separation in order to enhance the lifetime of charge carriers and also to increase interfacial charge transfer. In order to overcome the drawback of $TiO_2$, the rare earth (lanthanide) elements can also be used.

*Doped $TiO_2$ Systems*

Rare earth elements are ideal dopants for modifying the crystal structure, electronic structure, and optical properties of $TiO_2$ due to the 4f electronic configuration and spectroscopic properties [65,66]. Lanthanide ions could act as effective electron scavengers to trap the CB electrons from $TiO_2$ [67]. Xu et al. studied doping with rare earth ions including $La^{3+}$, $Ce^{3+}$, $Er^{3+}$, $Pr^{3+}$, $Gd^{3+}$, $Nd^{3+}$, and $Sm^{3+}$ in $TiO_2$, which improved photocatalytic activity in the degradation of nitrite. Wang et al. [68] also illustrated doping with lanthanide ions ($La^{3+}$, $Er^{3+}$, $Pr^{3+}$, $Nd^{3+}$, and $Sm^{3+}$), which improved the photoelectrochemical properties and increased the photocurrent response and the photon current conversion efficiency in the range of 300–400 nm.

## 9. Conclusions

Rapid growth in population and energy usage have resulted in increased environmental pollution and energy demands in recent years. In order to meet this demand, research on innovative and efficient methods of using renewable energy sources such as sunlight have become a necessity. In this review, we have discussed hydrogen as a fuel, its current industrial sources from fossil fuels, and its generation from the electrolysis of water. We then illustrated the mechanics (thermos, chemical, and otherwise) that allow for water splitting, specifically solar water splitting. Afterwards, we moved on to reviewing photocatalytic water splitting and the difference between the one-step and two-step processes. From here,

we concluded that one-step water splitting is more cost efficient and should be the focus of future research. We then discussed the criteria for ideal photocatalytic materials, including that the band positions should lie between 1.6 and 2.7 eV in order to use light in the visible spectrum, and the limitations with the current material systems. We finally focused on the $TiO_2$ systems and their potential and drawbacks as photocatalysts for water splitting.

It is suggested that the $TiO_2$ systems be further investigated as photocatalysts. Specifically, investigations including effectively decreasing the band gap of $TiO_2$ in order to absorb light in the visible spectrum. This will increase the efficiency of $TiO_2$ as a photocatalytic material and will further research in the areas of hydrogen fuel from a renewable energy source.

**Author Contributions:** Methodology, investigation, and writing—original draft preparation, F.M.; writing—review and editing, T.G.; conceptualization, supervision, and funding acquisition, P.-I.G. All authors have read and agreed to the published version of the manuscript.

**Funding:** This research was funded by the National Science Foundation, grant number CMMI-1833345.

**Conflicts of Interest:** The authors declare no conflict of interest.

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
