# Peer review of "Photochemical Water Splitting via Transition Metal Oxides"

_catalysts, doi:10.3390/catal12111303_

Round 1
Reviewer 1 Report
The authors provided a review on photocatalytic water splitting, focusing on mechanisms, desired material properties, and literature examples. There are some areas that need improvement:
1. As a review, the number of references seem inadequate, at only 67. In particular, it can be seen that Table 2 does not even cover all the possible catalysts shown in Fig 6. A more comprehensive literature review should be carried out.
2. Equation 1: it should be a forward arrow rather than equal sign.
3. Equation 4: it seems odd to write the potential of an oxidation reaction as more negative than that of a reduction reaction. Contrast with Fig 2, where this is positive.
4. Equation 6: the overall water splitting potential should just be 1.23 V (ie. the potential to apply across the 2 electrodes), not -1.23V vs NHE.
5. The fonts in Fig 6 are too small and hard to read; the text should be enlarged.
6. The authors should suggest, as recommendations for future research, specific materials systems for investigation for use in photocatalyst application.
Author Response
Dear Reviewer,
Thank you very much for your thorough review of our manuscript. We believe we have addressed your concerns in the following list:
- Our research generally focuses on transition metal oxides, which is why the materials systems may seem more limited than perhaps you were expecting. In order to maintain the integrity of the review article, the title has been revised to "Photochemical Water Splitting via Transition Metal Oxides" to better describe the scope of the manuscript to readers.
- The equals sign in Equation 1 has been revised to a forward arrow.
- The potential of the oxidation reaction in Equation 2 has been revised to be positive to be in agreement with Figure 2.
- The overall water splitting potential has been revised to simply "1.23 V" as opposed to "-1.23 V vs NHE".
- The text in Figure 6 has been enlarged from 7 and 8 point, to 9 and 10 point, respectively. This change should allow for easier reading of the values. The Figure has also been enlarged to use same width as body text.
- As the manuscript concludes with TiO2, a few sentences have been added to the conclusion in order to suggest further investigation of the TiO2 system for photocatalysis research.
Thank you again for your review of our manuscript. Please let us know if you have any other concerns.
Sincerely,
The Authors
Reviewer 2 Report
Title: Photochemical Water Splitting
Authors: Fateh Mikaeili, Tessa Gilmore, Pelagia-Iren Gouma
In this paper, Mikaeili and coworkers review the recent progress in water splitting capabilities by the means of the photocatalysis. The structure of the article fulfills the structure of a research article. The manuscript is well organized, very interesting for the increasing community working on renewable energy sources applications and particularly profiting of the specific characteristics of the photochemical water splitting. This is a good review, very simplified and easy to read and I recommend the publication after minor revision.
In my opinion, there is only few minor points that should be considered for optimizing the manuscript as follows:
- Please check the lines 141-142, page 4: the oxygen is oxidized by the holes in order to produce O2;
- Please add the reference regarding the confirmation of the differences between the free energy of rutile and anatase confirmed by Gouma et al. at page 11, line 405.
Author Response
Dear Reviewer,
Thank you very much for your kind words regarding our review manuscript. We have revised the document in accordance with your comments and have included the following:
- The H2 in line 141-142 has been revised to O2 to correctly state "... water reacts with the charge carriers in two separate redox reactions and therefore is reduced by electrons to produce H2 in the same time its oxidized by holes to produce O2."
- A reference has been added to the reference list and to line 405-406 regarding the confirmation of the differences between the free energy of rutile and anatase confirmed by Gouma et al. (New reference [58]).
Again, thank you very much for your kind review and suggestion to publish.
Sincerely,
The Authors
Round 2
Reviewer 1 Report
I am happy to accept the manuscript in its current state.